# LGCCT: A Light Gated and Crossed Complementation Transformer for Multimodal Speech Emotion Recognition

**DOI:** 10.3390/e24071010

**Published:** 2022-07-21

**Authors:** Feng Liu, Si-Yuan Shen, Zi-Wang Fu, Han-Yang Wang, Ai-Min Zhou, Jia-Yin Qi

**Affiliations:** 1Institute of AI for Education, East China Normal University, Shanghai 200062, China; 52205901024@stu.ecnu.edu.cn; 2School of Computer Science and Technology, East China Normal University, Shanghai 200062, China; 51215901045@stu.ecnu.edu.cn (S.-Y.S.); 51215901028@stu.ecnu.edu.cn (H.-Y.W.); 3School of Computer Science, Beijing University of Posts and Telecommunications, Beijing 100876, China; fuziwang@bupt.edu.cn; 4Shanghai Key Laboratory of Mental Health and Psychological Crisis Intervention, School of Psychology and Cognitive Science, East China Normal University, Shanghai 200062, China; 5Institute of Artificial Intelligence and Change Management, Shanghai University of International Business and Economics, Shanghai 200062, China

**Keywords:** entropy invariance, multimodal speech emotion recognition, cross-attention, gate control, lightweight model, computational affection

## Abstract

Semantic-rich speech emotion recognition has a high degree of popularity in a range of areas. Speech emotion recognition aims to recognize human emotional states from utterances containing both acoustic and linguistic information. Since both textual and audio patterns play essential roles in speech emotion recognition (SER) tasks, various works have proposed novel modality fusing methods to exploit text and audio signals effectively. However, most of the high performance of existing models is dependent on a great number of learnable parameters, and they can only work well on data with fixed length. Therefore, minimizing computational overhead and improving generalization to unseen data with various lengths while maintaining a certain level of recognition accuracy is an urgent application problem. In this paper, we propose LGCCT, a light gated and crossed complementation transformer for multimodal speech emotion recognition. First, our model is capable of fusing modality information efficiently. Specifically, the acoustic features are extracted by CNN-BiLSTM while the textual features are extracted by BiLSTM. The modality-fused representation is then generated by the cross-attention module. We apply the gate-control mechanism to achieve the balanced integration of the original modality representation and the modality-fused representation. Second, the degree of attention focus can be considered, as the uncertainty and the entropy of the same token should converge to the same value independent of the length. To improve the generalization of the model to various testing-sequence lengths, we adopt the length-scaled dot product to calculate the attention score, which can be interpreted from a theoretical view of entropy. The operation of the length-scaled dot product is cheap but effective. Experiments are conducted on the benchmark dataset CMU-MOSEI. Compared to the baseline models, our model achieves an 81.0% F1 score with only 0.432 M parameters, showing an improvement in the balance between performance and the number of parameters. Moreover, the ablation study signifies the effectiveness of our model and its scalability to various input-sequence lengths, wherein the relative improvement is almost 20% of the baseline without a length-scaled dot product.

## 1. Introduction

Emotion plays a key role in interpersonal communication [1], wherein not only linguistic messages but also acoustic messages convey individual emotional states. In many areas, such as human–computer interaction (HCI) [2], healthcare and cognitive sciences, much emphasis has been placed on developing tools to recognize human emotion in vocal expressions [3]. Recent booming advances in deep learning also promote the development of emotion recognition [4], a research field enabling machines to identify human emotion. Meanwhile, application requirements push the progress of lightweight models with high performance.

Focusing on acoustic features, a number of works have contributed to improving the performance of speech emotion recognition. Based on low-level descriptors (LLDs) in short frames, acoustic feature representations are extracted by deep learning networks, such as a convolutional neural network (CNN) [5], recurrent neural network (RNN) [6], etc. Some variant module architectures like CNN-LSTM [7], have also been developed to extract feature sequences and capture temporal dependencies.

Undoubtedly, linguistic information and acoustic information matters to speech emotion recognition [8]. Thus, both textual modality and audio modality should be taken into account to accomplish the task of multimodal emotion recognition. For audio modality, the process of feature extraction resembles that in unimodal speech emotion recognition. For textual modality, word-embedding models like GloVe [9] are commonly utilized.

What makes multimodal emotion recognition more challenging than unimodal emotion recognition is the process of modality fusion. Some early works concatenate different features as the input to the deep neural network [10]. To fuse the modalities in a deeper level, the standard transformer architectures [11] are widely extended to aggregate knowledge from one modality to the other, and, in this way, the learned modality-fused representation is enhanced [12,13].

Notwithstanding improvements made by prior works, the proportion of the modality-fused representation is seldom considered. To tackle this problem, we apply the gate-control mechanism [14] to the cross-attention module to decide whether to keep the source modality information or override the target modality information. 

Most of the high performance of existing models are dependent on a great number of learnable parameters [15,16], ignoring the potential application in some promising areas like human–computer interaction (HCI), which requires real-time and light models. Thus, a lightweight model is necessary to improve the feasibility and practicability of the application of speech emotion recognition. Additionally, the transformer may have difficulty generalizing to a sequence with a different length than the fixed ones while training, which impair the performance under actual HCI scenarios. To handle this problem, we follow prior works [17], just multiplying attention logits by a hyperparameter, and justify it from a view of entropy.

In this paper, we propose LGCCT, a lightweight gated and crossed complementation transformer for multimodal speech emotion recognition. First, we utilize CNN-BiLSTM and BiLSTM [18] to extract audio features and textual features, respectively. Then the cross-attention module reinforces one modality feature with the other, and the gate mechanism functions as a flow control unit to balance the proportion of the two modalities and the length-scaled dot-product operation enhance the generalization to unseen sequence length. Finally, the fully connected layers followed by the transformer encoder layers predict the emotion. 

Our contribution can be summarized as follows.

We propose a model to fuse the text-modality and audio-modality representation and learn the mapping from modality-fused representation to emotion categories.We adopt length-scaled attention module to improve the performance of the model when applied to various testing sequence length and theoretically interpret the determination of the scaled hyperparameter from a view of entropy.We apply a gate-control mechanism to the traditional cross-attention module. The effectiveness is verified by the ablation study.We reach a balance between the performance and the number of parameters (only 0.432M). Experiments are conducted on the CMU-MOSEI dataset [19]. The experiments also prove the generalization of our model to unseen sequence length. Compared with the baseline without a length-scaled dot product, the relative improvement is about 20%.

## 2. Related Works

Some early works for unimodal speech emotion recognition use traditional machine learning methods, such as a hidden Markov model [20], decision tree [21], and support vector machines [22]. With the development of deep learning methods, deep neural network (DNN)-based models in speech emotion recognition have thrived, like convolutional neural networks (CNN), recurrent neural networks (RNN) and long-short-term memory (LSTM) networks [6,7]. Some early works construct utterance-level features from segment-level probability distributions, and the extreme learning machine learns to identify utterance-level emotions [23]. Ref. [24] proposes a DNN-decision tree SVM model to extract the bottleneck features from confusion degree of emotion. CNNs mostly use spectrograms or audio features such as mel-frequency cepstral coefficients (MFCCs) and low-level descriptors (LLDs) as the inputs, followed by fully connected layers to predict the emotions [25]. RNN- and LSTM-based models take the temporal features into consideration and tackle this problem through sequence modeling [26]. Hybrid models like CNN-BiLSTM have also been adopted to effectively learn the information that represents emotions directly from conventional audio features [7,27]. Recently, the attention-based models and transformers have made significant progress in a range of fields [28,29]. Attention modules are used to learn the short-time frame-level acoustic features that are emotionally relevant, so that the temporal aggregated features can serve as more discriminative representation for classification [6]. Ref. [28] incorporates muti-task learning with attention-based hybrid models to better represent emotion features.

Emotion recognition in natural language processing (NLP) is also called sentiment analysis [30]. Early works take as input word embeddings, such as GloVe [9] and word2vec [31]. RNNs are capable of encoding the relations between sentences and capturing semantic meaning to distinguish sentiment better [32]. TextCNN [33] is a well-known convolutional neural network for sentence classification and is also widely applied to sentiment analysis [34]. The idea of attention is also popular in NLP. Ref. [21] uses a 2-D matrix to represent the embedding and introduces self-attention to extract an interpretable sentence embedding. In recent years, transformer-based self-supervised pretrained models, like BERT, thrive in NLP [11,35]. An increasing number of works take pretrained models as an encoder and get great performance boost [36,37].

Considering the fact that audios are composed of not only speech but also textual content, which can be extracted from the audio-based data, multimodal approaches using acoustic and lexical knowledge have also been explored. To further improve the accuracy, approaches that fuse audio, video and text are also a hot topic. There are mainly three kinds of future fusion strategies: attention-based feature fusion, GNN-based feature fusion [38,39,40] and loss-function-based feature fusion. For attention-based strategies, Ref. [18] proposes the bi-bimodal fusion network (BBFN) that performs fusion and separation on pairwise modality representations. Ref. [41] combines multi-scale CNN, statistical pooling unit and an attention module to exploit both acoustic and lexical information from speech. Ref. [13] proposes a multimodal transformer with the cross-modal attention mechanism to address this problem in an end-to-end manner. With such an idea, Ref. [42] uses both cross-modal attention and self-attention to propagate information within each modality. Ref. [43] designs a novel sparse transformer block to relieve the computational burden. Refs. [44,45] do the feature-fusion task by transferring it to a bi-modal translation task. For GNN-based strategies, Ref. [46] uses GCN to explore a more effective way of utilizing both multimodal and long-distance contextual information. For loss-function-based strategies, Ref. [47] hierarchically maximizes the mutual information in unimodal input pairs and between the multimodal fusion result and the unimodal input in order to maintain task-related information through multimodal fusion.

However, these methods ignore the fact that speech emotion recognition is needed mostly for real-time applications. Besides improving the accuracy by stacking models and arithmetic power, other factors such as being lightweight and showing efficiency and scalability to testing sequences with unfixed lengths are also necessary for practical applications. Thus, we will focus on reducing the number of parameters and improving the generalization to different testing sequences while maintaining performance.

## 3. Methodology

In this section, we will introduce the architecture of our network shown in Figure 1. The audio sequences are encoded by CNN-BiLSTM, while the text sequences are encoded by BiLSTM. Then the cross-attention module is utilized to fuse one modality representation into another modality representation respectively. The integration of the original-modality and the fused-modality representation is then controlled by the retain gate and the compound gate. The length-scaled coefficient is introduced while calculating the attention matrix to improve the generalization to different lengths, the validity of which will be illustrated from a view of entropy. To enhance the feature representation, the stacked transformer encoder is followed, and the classification is completed by the fully-connected layers. We will then illustrate our model in detail.

### 3.1. Text Encoder

We denote the text sequence as Xt={x1t,x2t,⋯,xnt}∈ℝn×dt, where dt denotes the word embedding dimension and n denotes the length of the sequence. BiLSTM is applied to capture the textual contextual dependencies.
(1)Ht=BiLSTM(Xt)
where Ht={h1t,h2t,⋯,hnt}∈ℝn×2dt′; Ht is the encoded feature representation, and dt′ is the dimension of the hidden states.

### 3.2. Audio Encoder

We denote the audio sequence as Xa={x1a,x2a,⋯,xna}∈ℝn×da, where da denotes the dimension of low-level acoustic features and n denotes the length of the sequence. For convenience, we set the length of the audio sequence equal to that of the text sequence (namely, da is equal to dt). Convolution layers are designed to extract high-level feature representation. Specifically, we use Conv1d to integrate the temporal information.
(2)HCNNa=Conv1d(Xa)

Then the BiLSTM takes Hcnna as input and outputs the audio contextual feature representation.
(3)Ha=BiLSTM(HCNNa)
where Ha={h1a,h2a,⋯,hna}∈ℝn×2da′ is the encoded feature representation, and da′ is the dimension of the hidden states.

### 3.3. Cross-Attention Module

As shown in Figure 1, we use the transformer encoder to generate modality-fused representation so that the two kinds of modality information are fused bidirectionally. Herein, we define the source-modality representation as HS∈ℝn×dS and the target modality representation as HT∈ℝn×dT, where {S,T}∈{t,a}. The process of modality fusion can be formulated as follows.
(4)Q=WQ×HT
(5)K=WK×HS
(6)V=WV×HS
where Q∈ℝn×dQ is the query matrix, K∈ℝn×dK is the key matrix’ V∈ℝn×dV is the value matrix, and × denotes matrix multiplication. Specifically, we set dQ, dK, dV equal to the dimension of target modality dT, denoted as d in the following. The source modality is transformed to the pair of key and value information while the target modality is transformed into the query information.

Then the fused-modality representation H′∈ℝn×dV is calculated by length-scaled dot-product attention.
(7)A=softmax(λQKTd)
(8)H′=AV
where softmax operation is applied to the dimension of sequence; A∈ℝn×n is the attention matrix, and λ is a hyperparameter to enable well length generalization, which will be illustrated in the next section.

Following the cross-attention module, layer normalization is designed to attend to original modality in the other modality.
(9)$$h^{S\to T}=LN\left(H^{\prime }+H^{T}\right)$$
where LN means layer normalization.

To aggregate the feature representation, a fully connected feed-forward network is utilized after the cross-attention module.
(10)$$H^{S\to T}=LN\left(h^{S\to T}+FFN\left(h^{S\to T}\right)\right)$$
where FFN means fully connected feed-forward network. The overview of cross-attention module is provided in Figure 2.

### 3.4. Entropy Invariance for Attention Operation

Following [17,48], we introduce a constant λ to improve the length generalization of attention operation, which can be interpreted from the perspective of entropy. In real-world applications, the length of an input sequence can be arbitrary, although the length is fixed during the training phase. The attention weight of the same token calculated in Equation (7) is supposed to converge to the same value independent of the length. From the view of entropy, we can consider the uncertainty as the degree of attention focus and revisit Equation (7) as follows.
(11)pij=eλqi·kj∑j=1neλqi·kj
(12)ℋi=−∑j=1npijlogpij
where qi∈ℝd is the i-th query vector in the input sequence; kj∈ℝd is the j-th key vector. qi·kj is the dot product of these two vectors, reflecting the similarity of the two vectors, and pij is the attention score between the i-th token and j-th token in the sequence with total length of n. It should be noted that pij is actually the element at row i and column in the attention matrix A. ℋi is the entropy of the i-th token. 

We can take the attention score as uncertainty. Specifically, the entropy is zero when the attention is attended to only one token, and the entropy is logn when the attention is distributed uniformly. Then we provide an approximate theoretical justification for the determination of hyperparameter λ. Equation (12) can be rewritten as:(13)$$\begin{array}{c} \;{ℋ}_{i}=\log\overset{n}{\underset{j=1}{\sum }}e^{\lambda q_{i}\cdot k_{j}}-\lambda \overset{n}{\underset{j=1}{\sum }}p_{ij}e^{\lambda q_{i}\cdot k_{j}}=\log n+\log\frac{1}{n}\overset{n}{\underset{j=1}{\sum }}e^{\lambda q_{i}\cdot k_{j}}-\lambda \overset{n}{\underset{j=1}{\sum }}p_{ij}e^{\lambda q_{i}\cdot k_{j}} \end{array}$$

Here the second term can be approximated as:(14)$$\log\frac{1}{n}\overset{n}{\underset{j=1}{\sum }}e^{\lambda q_{i}\cdot k_{j}}\approx \log\exp\left(\frac{1}{n}\overset{n}{\underset{j=1}{\sum }}\lambda q_{i}\cdot k_{j}\right)=\log\lambda \bar{q_{i}\cdot k_{j}}\;\;$$

Based on the hypothesis that the softmax operation can be used as a continuous, differentiable approximation to argmax [49], the third term can be approximated as:(15)$$\lambda \overset{n}{\underset{j=1}{\sum }}p_{ij}e^{\lambda q_{i}\cdot k_{j}}\approx \lambda \underset{1\leq j\leq n}{\max}\left(e^{\lambda q_{i}\cdot k_{j}}\right)$$

Therefore, we have:(16)$${ℋ}_{i}\approx \log n-\lambda \left(\underset{1\leq j\leq n}{\max}\left(e^{\lambda q_{i}\cdot k_{j}}\right)-\log\lambda \bar{q_{i}\cdot k_{j}}\right)$$

To mitigate the influence of the sequence length *n*, λ ∝ logn. For convenience, we set λ as logn. Equation (7) can be redefined as follows:(17)$$A=softmax\left(\frac{\log nQK^{T}}{\sqrt{d}}\right)$$

In this way, the contribution of the input token is more stable so that the attention matrix is theoretically more robust to the variation of input length. We determine the value of λ, and the next procedure is the same as mentioned above.

### 3.5. Gate Control

Based on the idea of close and open access of information flow [14], the gate unit is introduced to our network architecture. Some of the original-modality information should be attended to the fused-modality information.
(18)HS→T=HS→T×Gi+HT×Gr
where Gi∈ℝd×d represents the learnable integrate gate, and Gr∈ℝd×d represents the learnable retain gate. With learnable weights, the integrate gate decides how much fused information should be combined, and the retain gate decides how much original information should be preserved.

### 3.6. Classification

The transformer encoder layer is then employed, taking the concatenation of bidirectional modality information as input, as shown in Figure 3. Eventually, the classification is performed by fully connected feed-forward network.
(19)$$\hat{X}=FFN\left(Transformer\left(\left[H^{S\to T},H^{T\to S}\right]\right)\right)$$
where [∙,∙] denotes the concatenation of bidirectional modality information; transformer denotes the transformer encoder layer, and $\hat{X}$ denotes the predicted emotion category.

## 4. Results

In this section, we evaluate our model on CMU-MOSEI [19]. The implementation details and the experimental results are illustrated in this part.

### 4.1. Dataset and Metrics

CMU-MOSEI is a human multimodal sentiment-analysis dataset consisting of 23,453 sentences from YouTube videos, involving 1000 distinct speakers and 250 topics. The gender in the dataset is balanced (57% male to 43% female). The average length of sentences is 7.28 s, and acoustic features are extracted at a sampling rate of 20 Hz. Each sample is labeled by human annotators with a sentiment score from −3 to 3, including highly negative, negative, weakly negative, neutral, weakly positive, positive and highly positive.

The train/validation/test splits are provided by the CMU Multimodal Data SDK, wherein the same speaker does not appear in both train and test splits to ensure speaker independency. The length of the aligned sequences is 50. Using P2FA [50], the audio stream is aligned with the word along the timestep, within which the two modality features are averaged. For the text modality, the transcripts are processed with pre-trained GloVe [9] word embeddings, and the embeddings are 300-dimensional vectors. For the audio modality, the low-level 74-dimension vectors are extracted by COVAREP [51], including 12 Mel-frequency cepstral coefficients (MFCCs), pitch tracking and voiced/unvoiced segmenting features, glottal source parameters, peak slope parameters and maxima dispersion quotients. 

Consistent with the previous works [12,52], we adopt the metrics of 7-class accuracy (from strongly negative to strongly positive), binary accuracy (positive/negative sentiments) and F1 score (harmonic mean of the binary precision and recall). Specifically, the predicted digit will be rounded first. For 7-class accuracy, the predicted digit will be compared with the annotated sentiment score from −3 to 3. For binary accuracy, the predicted digit will be classified to positive or negative sentiment according to its positivity and negativity. The F1 score is the harmonic mean of the binary precision and recall.

### 4.2. Implementation Details

Our LGCCT is implemented by Pytorch [53] and optimized by Adam [54], with a learning rate 1 × 10^−3^, 40 epochs, a decay rate of 0.1 after 20 epochs, batch size of 24, a gradient clip of 1.0 and an output dropout rate of 0.1. The other hyperparameter mentioned in Section 3 are shown in Table 1. The dimensions of the hidden states H in the transformer are unified to 30.

The hardware environment for running is as follows: CPU: Intel(R) Xeon(R) Silver 4210R @ 2.40 GHz; GPU: NVIDIA Quadro RTX 8000; system running environment: Ubuntu 18.04.6.

### 4.3. Baselines

EF-LSTM. Early fusion LSTM concatenates the inputs from different modalities as the input to a single LSTM and classifies the feature vectors.

LF-LSTM. Late fusion LSTM describes each modality information separately, and the fusion takes place at the decision level.

RAVEN [55]. The proposed recurrent attended variation embedding network is composed of three parts, including nonverbal subnetworks, gated modality-mixing network and multimodal shifting.

MCTN [56]. The cyclic translation mechanism based on RNN is designed to learn joint representations.

MulT [12]. This model uses the cross-modal transformer, namely a deep stack of several cross-modal attention blocks, to fuse different modalities.

MISA [16]. Based on LSTM and pretrained BERT [35], MISA projects each modality to two subspaces.

BBFN [15]. The BERT encoder is utilized to obtain feature representation, which is then fused by transformer-like modules.

### 4.4. Comparison with Baseline Models

As shown in Table 2 and Figure 4, our model shows superiority in the balance between parameters and performance and maintains the higher F1 score than almost all models with limited number of parameters. $Acc_{7}$ denotes 7-class accuracy; $Acc_{2}$ denotes binary accuracy, and F1 denotes the F1 score. In terms of performance, our model outperforms the other models in $Acc_{2}$ and F1 score, except for BBFN, which utilizes the BERT encoder, requiring a large number of trained parameters and memory space. It is noteworthy that the axis of parameters in Figure 4 omits the range from 1.4 M and 110.4 M, and, thus, the required parameters for BBFN and MISA are huge. Considering the balance between parameters and performance, the model scale is enlarged by more than 110 M parameters for about 4% absolute performance improvement, indicating the imbalance tradeoff between performance and computational complexity. Besides, in almost all metrics, our model performs better than the three models trained with tri-modality information, learning richer modality information. In terms of the number of parameters, our model ranks only second to MCTN. However, the $Acc_{2}$ and F1 scores of our model are much higher than that of MCTN, by more than 2%, while the $Acc_{7}$ is slightly worse than that of MCTN by 0.14%. Although the parameters of LSTM-based models are low, performance is also limited, and our LGCCT surpass them on almost all of the metrics. This is partly due to the fact that they do not take into account the interactions between the modalities, just concatenating the modalities. Our model adopts efficient modules, such as cross-attention and a gate-control mechanism, to fuse the modality information and maintain the balance between the source modality and target modality. The experiments indicate the effectiveness of our model.

### 4.5. Ablation Study

To study the effect of different parts on the performance, we conduct some experiments on the CMU-MOSEI dataset. The results are shown in Table 3. First, we evaluate the influence of the gate-control mechanism. The $Acc_{7}$ drops about 4.6%, indicating the effectiveness of the introduced gate units. Second, the audio encoder CNN-BiLSTM and text encoder BiLSTM are removed. The LGCCT model outperforms these two models in all metrics, suggesting the importance of feature extraction. It is noteworthy that the performance degrades most when the feature encoder is removed, signifying the fact that the feature encoder aggregates the original modality information and that the representation to the modality-fusing modules is powerful. Finally, the transformer encoder ahead of the fully connected layer is removed. The results show that it is necessary to apply the self-attention module to encode the modality-fused representation. However, the performance without a transformer degrades least, but the parameters are cut down most in the ablation study. We assume that the cross-attention operation in the modality-fusing module manages to attend to interactions between multimodal sequences, and, thus, the contribution of the last transformer to the performance is restricted. The ablation study implies the function of the components of our model and verifies the contribution of each module to the performance.

### 4.6. Length-Scaled Attention

To mitigate the problem that the length of the training sequence is fixed while the length of the testing sequence may vary, we introduce length-scaled attention. For the standard input, the length of the training sequence and the testing sequence are unified to 50, which is referred to in Table 4 and Table 5. To validate the effectiveness of length-scaled attention, we clip the original sequence according to two proportions of the original length: 80% and 60% of the original one. This experiment configuration yields two other length: 40 and 30, respectively, denoted in Table 4 and Table 5. Then we test/train the variant of LGCCT with/without length-scaled attention. Other configurations are kept default. Table 4 shows great improvement of the model when the length of the training sequence and the testing sequence varies, especially at a length of 30. To be specific, when training and testing on data with the same length, the effect of length-scaled attention is not obvious but length-scaling outperforms its counterpart by 12.9% and 6.9%, respectively, when training and testing on different lengths. A closer look at the result with a length of 30 shows that the model without length-scaled attention performs poorly when the testing sequence length is 50 but the training one is clipped to 30. For the binary accuracy in Table 4, the length-scaled dot product brings the relative improvement of 20%. A similar improvement is also shown in training all test settings with clipped length of 30, wherein the relative improvement is 11% on accuracy and 30% on the F1 score. Moreover, length scaling helps stabilize performance on the F1-score as shown in Table 5, while shorter testing sequences lead to serious performance degradation for vanilla attention operation, like a 57.8% F1 score when testing sequence is cut to 30. Interestingly, the model with length scaling does not show superiority over its counterpart without that, which to some extent reveals the data efficiency of our LGGCT when the length gap between the training and testing sequence is not large. We hypothesize such stable performance occurs because the length of text modality and audio modality is not always identical and is forced to be aligned, wherein sometimes zero logits are padded to the end of the sequence. This suggests that clipped data may compose of useless zero frames. Furthermore, the variant LGCCT with length-scaled attention manages to generalize to the sequence with a length different from the training set.

## 5. Conclusions

In this paper, we propose LGCCT, a lightweight gated and crossed complementation transformer for multimodal speech emotion recognition. Text encoder BiLSTM and audio encoder CNN-BiLSTM are utilized to obtain modality feature expression. At the heart of LGCCT, cross-attention modules fuse the modality information with each other, and the learnable gate-control mechanism controls the information flow and stabilizes the training process. Moreover, we apply length scaling to the attention module to improve the generalization of the transformer to various testing strings, which can be elaborated from a view of entropy invariance. In particular, the attention weight of the same token in the attention matrix is supposed to converge to the same value independent of the length. From the view of entropy, we can consider the uncertainty as the degree of attention focus. The attention scores can be consistent with the length of the input length just by multiplying the hyperparameter λ. At the top of the model, the fully connected forward network followed by the transformer encoder learns the mapping from modality-fused representation to emotion categories. In the experiments, we compare our model with baseline models on the benchmark dataset and further the ablation study. Our model achieves the balance between performance and the number of parameters with only 0.432 M parameters. The results also show the effectiveness of each component, underlying the performance and lightweight footprint of our model. Furthermore, the length-scale attention does help the model generalize to various sequence lengths under the experiment with different sequence lengths.

From the view of multimodal speech emotion recognition, our method has shown a balance between performance and the number of parameters. We attribute this to two factors. First, we adopt efficient modules in our network, such as cross-attention and a gate-control mechanism. In this way not only can the inter-modality information communicate and mingle with each other, but the generated modality-fused information can also maintain the balance between the target modality and the source modality. Similar efficiency is maintained within the process of feature extraction. Second, we keep the dimension of the hidden states low. Just as shown in Table 1, all of the dimension are kept approximately 30, except for the input channels. In contrast, many previous works adopting the BERT encoder [35] have to adjust the input channel to 512 [15,16], far more than ours. It is noteworthy that we do not apply any other down-sampling after the feature-extraction stage, keeping the embedding dimension low and the sequence length the same. This design is similar to the classic ideas in computer vision, namely maintaining large activation maps while decreasing the quantity of parameters [57]. For a wide neural network, not all of the information contained in high-dimensional vectors are useful [58], and thus our LGCCT is designed as a narrower network so that the information can be more compact.

From the view of information theory, our entropy-based LGCCT variant is capable of generalizing to testing sequences with various lengths. We consider the degree of attention focus as the uncertainty and let the same token converge to the same value independent of the length by multiplying the predefined constant λ. Since the attention matrix is computed by the learnable parameters, the model is supposed to learn the value ideally. However, our experiment shows that the model with length-scaling performs more stably, while the model without length scaling fluctuates in performance. This phenomenon indicates that the perspective of entropy really works, and the inductive bias [59] can help the model find better solutions. Actually, the idea of entropy is widely applied to deep learning methods. One of the most-typical applications is cross entropy. This criterion serves as a loss function to compute the loss between input and target, especially when handling a classification problem with multiple classes [60]. More generally, other classic perspectives in information theory have been used in deep learning methods [47,61,62].

In the future, we will design a more effective gate mechanism, following some gate units such as LSTM cell and GRU gates [63]. Furthermore, other modalities like video can be considered, so that a tri-modal emotion recognition network can be developed for application in realistic scenarios. The effectiveness of length-scaled attention for multimodal emotion recognition may shed light on the wider usage of entropy, as well as information theory, in the deep learning community.

## Figures and Tables

**Figure 1 entropy-24-01010-f001:**
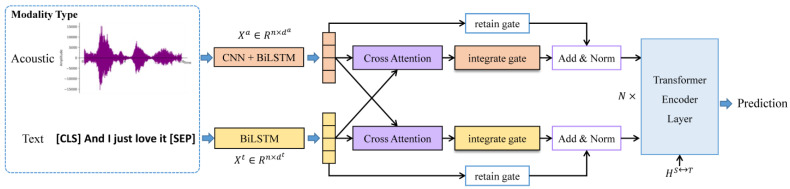
The overall architecture of LGCCT. CNN−BiLSTM and BiLSTM extract acoustic features and text features respectively. At the heart of the model, the cross−attention module with a gate−control mechanism fuses the modality information. The transformer encoder layers reinforce the modality-fused representation.

**Figure 2 entropy-24-01010-f002:**
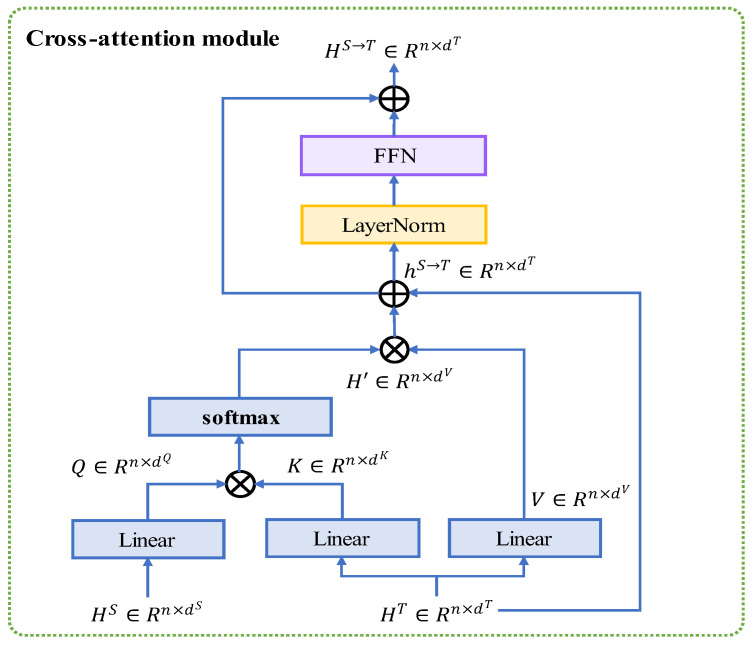
Cross-attention module fuses the modality information.

**Figure 3 entropy-24-01010-f003:**
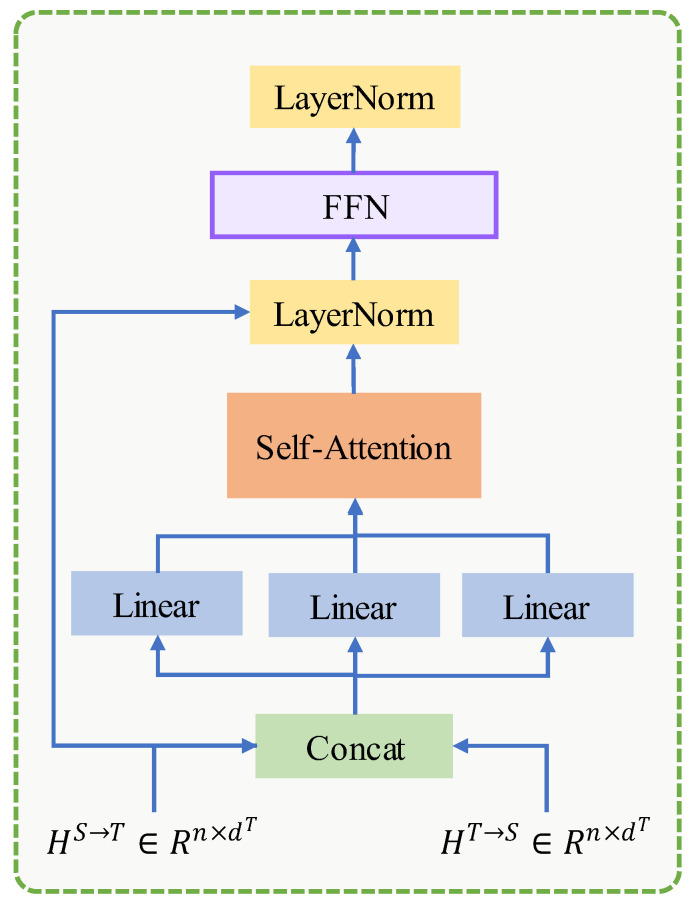
The architecture of the Transformer.

**Figure 4 entropy-24-01010-f004:**
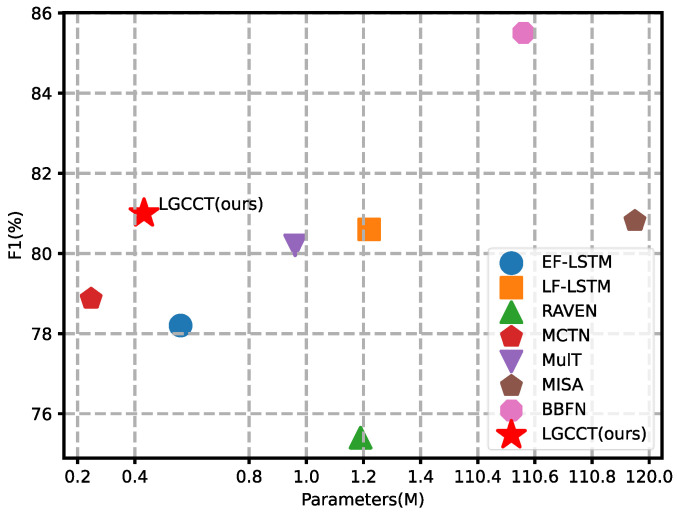
Comparison of the $F_{1}$ score of different models on CMU-MOSEI. The proposed LGCCT achieves the best performance with an order of magnitude smaller model size.

**Table 1 entropy-24-01010-t001:** Detailed dimensions of LGCCT.

Notation	Meaning	Value
n	Aligned input-sequence length	50
$d^{t}$	Word-embedding dimension	300
$d^{a}$	Audio feature dimension	74
$d^{t\prime }$	Encoded text feature dimension by BiLSTM	32
$d^{a\prime }$	Encoded audio feature dimension by CNN-BiLSTM	32
d	Hidden state dimension	30
λ	Length-scale logits	log50

**Table 2 entropy-24-01010-t002:** The performance and the number of parameters on the CMU-MOSEI dataset.

Method	#Params(M)	$Acc_{7}\left(\%\right)$	$F_{1}\left(\%\right)$	$Acc_{2}\left(\%\right)$
MulT	0.961	48.2	80.2	79.7
MCTN	0.247	47.64	78.87	77.86
MISA **	110.915	53.31	80.81	80.26
BBFN **	110.548	51.7	85.5	85.5
EF-LSTM *	0.56	47.4	78.2	77.9
LF-LSTM *	1.22	48.8	80.6	80.6
RAVEN *	1.19	45.5	75.4	75.7
LGCCT (ours)	0.432	47.5	81.0	81.1

* with tri-modality, namely audio, video and text. ** with pretrained BERT.

**Table 3 entropy-24-01010-t003:** Ablation study on the CMU-MOSEI dataset.

Model	#Params(M)	$Acc_{7}\left(\%\right)$	$Acc_{2}\left(\%\right)$	$F_{1}\left(\%\right)$
LGCCT	0.432	47.5	81.0	81.1
*w*/*o* gates	0.429	42.9	76.7	76.3
*w*/*o* CNN-BiLSTM & BiLSTM	0.354	40.9	70.7	70.8
*w*/*o* Transformer	0.203	40.3	75.6	78.0

**Table 4 entropy-24-01010-t004:** Accuracy comparisons on CMU-MOSEI with different length distributions.

	All = 50	Part = 30		Part = 40	
Type	Train All Test All	Train Part Test All	Train All Test Part	Train Part Test All	Train All Test Part
Length scaled	80.8	75.7	65.2	74.4	67.7
*w*/*o* length scaling	81.1	62.8	58.3	77.0	71.9

**Table 5 entropy-24-01010-t005:** F1-score comparisons on CMU-MOSEI with different length distributions.

	All = 50	Part = 30		Part = 40	
Type	Train All Test All	Train Part Test All	Train All Test Part	Train Part Test All	Train All Test Part
Length scaled	80.7	76.2	75.3	76.8	74.8
*w*/*o* Length scaled	81.0	77.2	57.8	78.3	72.3

## Data Availability

Not applicable.

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
