# Peer review of "LGCCT: A Light Gated and Crossed Complementation Transformer for Multimodal Speech Emotion Recognition"

_entropy, 2022, doi:10.3390/e24071010_

Round 1

Reviewer 1 Report

The changes to the manuscript have produced a much improved piece of work

Author Response

We would like to thank you for your kind letter and for reviewers’ positive comments.

Reviewer 2 Report

This paper proposes a Light Gated and Crossed Complementation Transformer (LGCCT) for multimodal speech emotion recognition. The model used is capable of fusing modality information efficiently. The degree of attention focus can be considered as the uncertainty, and the entropy of the same token should converge to the same value independent of the length. Experiments are conducted on the benchmark dataset CMU-MOSEI. Compared to the baseline models, the proposed model achieves 81.0% F1 score with only 0.432M parameters. This show an improvement in the balance between performance and the number of parameters.

The paper is well organized and readable. I have only two suggestions, which are described below, to be considered to improve the paper.

Authors should provide information on how Acc2 and Acc7 are determined or at least provide a link to a reference.

The authors present the results in Table 2 and then present these results graphically in Figures 4 and 5. I believe this is a duplication of results, and these two figures are unnecessary. Above all, both figures do not bring any additional information as it is difficult to read the results from the chart.

I recommend that the paper should be accepted with minor revision.

Author Response

This manuscript is a resubmission of an earlier submission. The following is a list of the peer review reports and author responses from that submission.

Round 1

Reviewer 1 Report

This seems an interesting paper, focused on acoustic signal analysis and feature extraction. The mathematical side looks plausible and free of error. However, the manuscript has serious flaws that need to be addressed.

Suggestions and comments:

- Why are there different fonts and styles in the Author Name’s Section? Additionally, there are some typos (Capital letters) when it comes to the Affiliation section, i.e., it should be [Shanghai] with a Capital [S].

- The Abstract could be rewritten, in order to better describe the contents of the manuscript. Provide an introduction to the discussed subject, later on followed by a description of your laboratory stand (what was done an with the aid of what equipment/software), followed by a summary of obtained results.

- Several typos are present, e.g., [Re-cent] or [re-search] on page 1.

- The Related Works section is far too short, it needs to be extended. Look for other papers focused on speech/music signal analysis, perception of speech signals, etc., that were recently published in journals as well as conference proceedings.

- Mathematical symbols appearing in text should be written in a different manner than plain text, i.e., using italics, so that they are easily distinguishable.

- Some sub-chapters, e.g., 4.1, 4.2, etc., are too short to be a standalone one. Consider reorganizing some fragments of the manuscript.

- About the utilized signal samples – what was the original file format? What about the bitrate and sampling rate? Low long were they (in seconds or minutes)? Did they include only speech or mixed audio+speech signals? What about the lectors – were they male/female individuals? And how many of them were present? What kind of language did it include? What kind of dialect was it? Several important information are missing.

- Additionally, were the processed (by the Authors) signal samples altered in any way – resampling, changing the bitrate or original file format, etc.?

- What about the utilized software/hardware in your laboratory stand? What kind of equipment did you utilize? What kind of software (freely available commercial software? Open source? Or custom build?) was used? What about third-party libraries, toolboxes, etc.?

- How many iterations did it take? What about some kind of statistical analysis, including confidence intervals, etc.?

- The Research part is far too short and not convincing at all. It should be rewritten and extended.

- The Conclusions part is far too short and not convincing. What about open issues and future studies? Provide additional feedback as well as source of inspiration for other authors.

- The References section, taking into account the topic of this paper, is far too short. It requires extensive extention.

After reading the whole paper I am not sure about the aim as well as contents of the paper. It requires a major revision and extension before it can be considered and processed further.

Author Response

Dear editors and reviewers:

We would like to thank you for your kind letter and for reviewers’ positive and constructive comments concerning our article entitled “LGCCT: An Light Gated and Crossed Complementation Transformer for Multimodal Speech Emotion Recognition”. These comments are all valuable and helpful for improving our article.

We have studied reviewers’ comments carefully and tried our best to revise our manuscript according to the comments. If there are any other modifications we could make, we would like very much to modify them. We hope that our manuscript could be considered for publication in your journal. Thank you very much for your help.

Point-by-point responses to the reviewers are listed below this letter.

Thank you and best regards.

Institute: East China Normal University, China

E-mail: lsttoy@163.com

Thanks very much for your attention to our paper.

Sincerely yours,

Dr. Feng Liu

July 07, 2022

Author Response to Reviews of

LGCCT: An Light Gated and Crossed Complementation Transformer for Multimodal Speech Emotion Recognition

Review                                                                                 

Reviewer#1:

  1. Why are there different fonts and styles in the Author Name’s Section? Additionally, there are some typos (Capital letters) when it comes to the Affiliation section, i.e., it should be [Shanghai] with a Capital [S].

Author Response: Thank you for your advice. We apologize for our manuscript error. The fonts and styles of the author name have been unified and typos in affiliation section are corrected as follows:

Authors: Feng Liu 1, 2, 3, †, §, Si-yuan Shen2, §, Zi-wang Fu3, Han-yang Wang2, Ai-min Zhou1, 2,4 * and Jia-yin Qi5, *

Affiliation:

1    Institute of AI for Education, East China Normal University, Shanghai, China;

2    School of Computer Science and Technology, East China Normal University, Shanghai, China;

3   School of Computer Science, Beijing University of Posts and Telecommunications, China

4    Shanghai Key Laboratory of Mental Health and Psychological Crisis Intervention, School of Psychology and Cognitive Science, East China Normal University, Shanghai, China;

5    Institute of Artificial Intelligence and Change Management, Shanghai University of International Business and Economics, Shanghai, China;

*   Correspondence: amzhou@cs.ecnu.edu.cn; ai@suibe.edu.cn

    ORCID: 0000-0002-5289-5761;

  • Co-first author;

Reviewer#1:

  1. The Abstract could be rewritten, in order to better describe the contents of the manuscript. Provide an introduction to the discussed subject, later on followed by a description of your laboratory stand (what was done an with the aid of what equipment/software), followed by a summary of obtained results.

Author Response: Thanks for your careful review and comments. We now illustrate the aim of speech emotion recognition task and the results in abstract. Briefly, our contribution is mainly two folds. (1) We propose a lightweight model with gated mechanism and cross-attention to classify emotions efficiently with only 0.432M parameters. (2) Additionally, we adopt length scaled dot product to calculate the attention score, which can be interpreted from a view of entropy theoretically. Experiments validate its generalization to testing data with unfixed length as well as the balance between performance and the number of parameters. The latest version of abstract is:

Abstract: Semantic-rich speech emotion recognition has a high degree of popularity in a range of areas. Speech emotion recognition aims to recognize human emotional states from the utterances, containing both acoustic and linguistic information. Since both textual and audio patterns play essential roles in speech emotion recognition (SER) tasks, various works have proposed novel modality fusing methods to exploit text and audio signals effectively. However, most of the high performance of existing models are dependent on a great number of learnable parameters and they can only work well on data with fixed length. Therefore, how to minimize computational overhead and improve generalization to unseen data with various lengths while maintaining a certain level of recognition accuracy is an urgent application problem. In this paper, we propose LGCCT, a light gated and crossed complementation Transformer for multimodal speech emotion recognition. First, our model is capable of fusing modality information efficiently. Specifically, the acoustic features are extracted by CNN-BiLSTM while the textual features are extracted by BiLSTM. The modality-fused representation is then generated by the cross-attention module. We apply the gate control mechanism to achieve a balanced integration of the original modality representation and the modality-fused representation. Second, the degree of attention focus can be considered as the uncertainty and the entropy of the same token should converge to the same value independent of the length. To improve the generalization of the model to various testing sequence lengths, we adopt the length scaled dot product to calculate the attention score, which can be interpreted from a view of entropy theoretically. The operation of length scaled dot product is cheap but effective. Experiments are conducted on the benchmark dataset CMU-MOSEI. Compared to the baseline models, our model achieves 81.0% F1 score with only 0.432M parameters, showing an improvement in the balance between performance and the number of parameters. Moreover, the ablation study signifies the effectiveness of our model and its scalability to various input sequence lengths, where the relative improvement is almost 20% to the baseline without length scaled dot product.

Reviewer#1:

  1. Several typos are present, e.g., [Re-cent] or [re-search] on page 1.

Author Response: Thanks for your advice, we have revised the article and now all the typos with extra ‘-’ are corrected.

Reviewer#1:

  1. The Related Works section is far too short, it needs to be extended. Look for other papers focused on speech/music signal analysis, perception of speech signals, etc., that were recently published in journals as well as conference proceedings.

Author Response: Thank you for your helpful suggestions. We have extended the related work section, including traditional machine learning methods, speech emotion recognition for speech signals, sentiment analysis for text, and multimodal emotion recognition. The improved related work section is:

Related Work: Some early works for unimodal speech emotion recognition use traditional machine learning methods, such as hidden Markov model [20], decision tree [21], and support vector machines [22]. With the development of deep learning methods, deep neural networks (DNN) based models in speech emotion recognition have thrived, like convolutional neural networks (CNN), recurrent neural networks (RNN), long-short term memory (LSTM) networks [6,7]. Some early works construct utterance-level fea-tures from segment-level probability distributions and the extreme learning machine learns to identify utterance-level emotions [23]. [24] proposes a DNN-decision tree SVM model to extract the bottleneck features from confusion degree of emotion. CNNs mostly uses the spectrograms or audio features such as mel-frequency cepstral coefficients (MFCCs) and low-level descriptors (LLDs) as the inputs, followed by fully connected layers to predict the emotions [25]. RNN and LSTM based models take the temporal features into consideration and tackle this problem through sequence modeling [26]. The hybrid model like CNN-BiLSTM is also adopted to effectively learn the information that represents emotions directly from conventional audio features [7,27]. Recently, the attention-based models and Transformers have made significant progress in a range of fields [28,29]. Attention modules are used to learn the short-time frame-level acoustic features that are emotionally relevant, so that the temporal aggregated features can serve as more discriminative representation for classification [6]. [28] incorporates muti-task learning with attention-based hybrid models to better represent emotion features.

Emotion recognition in natural language processing (NLP) is also called sentiment analysis [30]. Early works take as input word embeddings, such as GloVe [9] and word2vec [31]. RNNs are capable of encoding the relations between sentences and capturing semantic meaning to distinguish sentiment better [32]. TextCNN [33] is a well-known convolution neural network for sentence classification and is also widely applied to sentiment analysis [34]. The idea of attention is also popular in NLP. [21] uses a 2-D matrix to represent the embedding and introduces self-attention to extract an interpretable sentence embedding. In recent years, Transformer-based self-supervised pretrained models, like BERT, thrive in NLP [11,35]. An increasing number of works take pretrained models as encoder and get great performance boost [36,37].

Considering the fact that audios are composed of not only speech but also textual content, which can be extracted from the audio-based data, multimodal approaches using acoustic and lexical knowledge have also been explored. To further improve the accuracy, approaches that fuse audio, video and text are also a hot topic. There are mainly three kinds of future fusion strategies: attention-based feature fusion, GNN-based feature fusion [38-40] and loss-function-based feature fusion. For attention-based strategies, [18] proposes the Bi-Bimodal Fusion Network (BBFN) that per-forms fusion and separation on pairwise modality representations. [41] combines multi-scale CNN, statistical pooling unit and an attention module to exploit both acoustic and lexical information from speech.    [13] proposes Multimodal Transformer with the cross-modal attention mechanism to address this problem in an end-to-end manner. With such idea, [42] uses both cross-modal attention and self-attention to propagate information within each modality. [43] designs a novel sparse transformer block to relieve the computational burden. [44,45] do the feature fusion task by transferring it to a bi-modal translation task. For GNN-based strategies, [46] uses GCN to explore a more effective way of utilizing both multimodal and long-distance contextual information. For loss-function-based strategies, [47] hierarchically maximizes the Mutual Information in unimodal input pairs and between multimodal fusion result and unimodal input in order to maintain task-related information through multimodal fusion

However, these methods ignore the fact that speech emotion recognition is needed mostly for real-time applications. Besides improving the accuracy by stacking models and arithmetic power, lightweight and scalability to testing sequence with unfixed se-quence is also necessary for practical applications. Thus, we will focus on reducing the number of parameters and improving the generalization to different testing sequence while maintaining the performance.

Reviewer#1:

  1. Mathematical symbols appearing in text should be written in a different manner than plain text, i.e., using italics, so that they are easily distinguishable.

Author Response: Thank you for your advice. Mathematical symbols in text now are written in the same manner as that in equation.

Reviewer#1:

  1. Some sub-chapters, e.g., 4.1, 4.2, etc., are too short to be a standalone one. Consider reorganizing some fragments of the manuscript.

Author Response: Thanks for your advice, we have extended 4.1 and 4.2 respectively. The details of the dataset and our implementation are added to these two sub-chapters.

Reviewer#1:

  1. About the utilized signal samples – what was the original file format? What about the bitrate and sampling rate? Low long were they (in seconds or minutes)? Did they include only speech or mixed audio+speech signals? What about the lectors – were they male/female individuals? And how many of them were present? What kind of language did it include? What kind of dialect was it? Several important information are missing.

Author Response: CMU-MOSEI is a human multimodal sentiment analysis dataset consisting of 23,453 sentences from YouTube videos, involving 1,000 distinct speakers and 250 topics. The unaligned CMU-MOSEI sequences are extracted at a sampling rate of 20 Hz and the average length of sentences is 7.28 seconds. We use the aligned dataset provided by the CMU Multimodal Data SDK and the length of the preprocessed aligned sequences is 50. The text modality is processed as word embeddings while the audio signal features are extracted by COVAREP. The gender in the dataset is balanced (57% male to 43% female). The related information about CMU-MOSEI dataset has been added to section 4.1 dataset:

Dataset: CMU-MOSEI is a human multimodal sentiment analysis dataset consisting of 23,453 sentences from YouTube videos, involving 1,000 distinct speakers and 250 topics. The gender in the dataset is balanced (57% male to 43% female). The average length of sentences is 7.28 seconds and acoustic features are extracted at a sampling rate of 20Hz. Each sample is labeled by human annotators with a sentiment score from -3 to 3, including highly negative, negative, weakly negative, neutral, weakly positive, positive and highly positive.

The train/validation/test splits are provided by the CMU Multimodal Data SDK, where the same speaker doesn’t appear in both train and test splits to ensure speaker independency. The length of the aligned sequences is 50. Using P2FA [50], the audio stream is aligned with the word along the timestep, within which the two modality features are averaged. For the text modality, the transcripts are processed with pre-trained GloVe [9] word embeddings and the embeddings are 300-dimensional vectors. For the audio modality, the low level 74-dimension vectors are extracted by COVAREP [51], including 12 Mel-frequency cepstral coefficients (MFCCs), pitch tracking and voiced/unvoiced segmenting features, glottal source parameters, peak slope parameters and maxima dispersion quotients.

Reviewer#1:

  1. Additionally, were the processed (by the Authors) signal samples altered in any way – resampling, changing the bitrate or original file format, etc.?

Author Response: Thanks for your advice. The original audio signal is sampled at 20 Hz. Words and audio in the aligned benchmark dataset are aligned at phoneme level using P2FA forced alignment model, provided by the CMU Multimodal Data SDK. Due to the alignment processing, the sample rate of the data is actually altered but the authors of CMU-MOSEI don’t mention the concrete sample rate in their website or paper.

Reviewer#1:

  1. What about the utilized software/hardware in your laboratory stand? What kind of equipment did you utilize? What kind of software (freely available commercial software? Open source? Or custom build?) was used? What about third-party libraries, toolboxes, etc.?

Author Response: Thanks for your advice. The LGCCT model is implemented by Pytorch. The hardware environment for running is as follows: CPU: Intel(R) Xeon(R) Silver 4210R @ 2.40GHz; GPU: 2 * NVIDIA Quadro RTX 8000; system running environment: Ubuntu 18.04.6. The information is added to the manuscript now.

Reviewer#1:

  1. How many iterations did it take? What about some kind of statistical analysis, including confidence intervals, etc.?

Author Response: Thanks for your advice. We follow the benchmark of the emotion recognition on CMU-MOSEI and evaluate the three metrics, i.e. 7-class accuracy, binary accuracy, and F1 score. The implementation details have been added to section 4.2 implementation details.

Implementation details: Our LGCCT is implemented by Pytorch [53] and optimized by Adam with a learning rate 1e-3, 40 epochs, a decay rate of 0.1 after 20 epochs and batch size of 24, gradient clip of 1.0 and The output dropout rate of 0.1. The other hyperparameter mentioned in Section 3 are shown in Table 1. The dimensions of the hidden states H in Transformer are unified to 30.

Table 1. detailed dimensions of LGCCT

Notation

Meaning

Value

Aligned input sequence length

50

Word embedding dimension

300

Audio feature dimension

74

Encoded text feature dimension by BiLSTM

32

Encoded audio feature dimension by CNN-BiLSTM

32

Hidden state dimension

30

Length scale logits

The hardware environment for running is as follows: CPU: Intel(R) Xeon(R) Silver 4210R @ 2.40GHz; GPU: NVIDIA Quadro RTX 8000; system running environment: Ubuntu 18.04.6.

Reviewer#1:

  1. The Research part is far too short and not convincing at all. It should be rewritten and extended.

Author Response: Thanks for your advice. More details are added and we conduct more experiments on various sequence length (including 100%, 80%, 60% of the original length). And the illustration is as follows.

Comparison: As shown in Table 2 and Fig. 2, our model shows superiority in the balance between parameters and performance.  denotes 7-class accuracy,  denotes binary accuracy and  denotes  score. In terms of performance, our model outperforms the other models in  and  score, except for BBFN, which utilizes BERT encoder requiring a large number of trained parameters and memory space. Considering the balance between parameters and performance, the model scale is enlarged by more than 110 M parameters for about 4% absolute performance improvement, indicating the imbalance tradeoff between performance and computational complexity. Besides, in almost all of metrics, our model performs better than the three models trained with tri-modality information, learning richer modality information. In terms of the number of parameters, our model ranks only second to MCTN. However,  and  score of our model is much higher than that of MCTN by more than 2% while the  is slightly worse than that of MCTN by 0.14%. Although the parameters of LSTM-based models are low, the performance is also limited and our LGCCT surpass them on almost all of the metrics. This is partly due to the fact that they do not take into account the interactions between the modalities, just concatenating the modalities. Our model adopts the efficient modules, such as cross-attention, gate control mechanism, to fuse the modality information and maintain the balance between the source modality and target modality. The experiments indicate the effectiveness of our model.

Ablation study: To study the effect of different parts on the performance, we conduct some experiments on CMU-MOSEI dataset. The results are shown in Table 3. First, we evaluate the influence of the gate control mechanism. The  drops about 4.6%, indicating the effectiveness of introduced gate units. Second, the audio encoder CNN-BiLSTM and text encoder BiLSTM are removed. The LGCCT model outperforms these two models in all metrics, suggesting the importance of feature extraction. It is noteworthy that the performance degrades most when the feature encoder is removed, signifying the fact that the feature encoder aggregates the original modality information and the representation to the modality-fusing modules is powerful. Finally, the Transformer encoder ahead of the fully connected layer is removed. The results show that it is necessary to apply self-attention module to encode the modality-fused representation. However, the performance without Transformer degrades least but the parameters are cut down most in the ablation study. We assume that the cross-attention operation in the modality-fusing module manages to attend to interactions between multimodal sequences and thus the contribution of the last Transformer to the performance is restricted. The ablation study implies the function of the components of our model and verifies the contribution of each module to the performance.

Length scaled attention: To mitigate the problem that the length of training sequence is fixed while the length of the testing sequence may vary, we introduce length scaled attention. For the standard input, the length of the training sequence and the testing sequence are unified to 50, which is referred to all In Table 4 and Table 5. To validate the effectiveness of length scaled attention, we clip the original sequence according to two proportion to the original length: 80% and 60% of the original one. This experiment configuration yields two other length: 40 and 30 respectively, denoted as part in Table 4 and Table 5. Then we test/train the variant of LGCCT with/without length scaled attention. Other configurations are kept default. Table 4 shows great improvement of the model when the length of the training sequence and the testing sequence varies, especially on the length of 30. To be specific, when training and testing on data with the same length, the effect of length scaled attention is not obvious but length scaling outperform its counterpart by 12.9% and 6.9% respectively when training and testing on different lengths. Let have a closer look at the result on the length of 30. The model without length scaled attention performs poorly when the testing sequence length is 50 but the training one is clipped to 30. For the binary accuracy in Table 4, length scaled dot product brings the relative improvement of 20%. Similar improvement is also shown in train all test part setting with clipped length of 30, where the relative improvement is 11% on accuracy and 30% on F1 score. Moreover, length scaling helps stabilize the performance on F1-score as shown in Table 5 while shorter testing sequences lead to serious performance degradation for vanilla attention operation, like 57.8% F1 score when testing sequence is cut to 30. Interestingly, the model with length scaled doesn’t show superiority over its counterpart without that, which to some extent reveals the data efficiency of our LGGCT when the length gap between training and testing sequence is not large. We hypothesize such stable performance occurs because the length of text modality and audio modality is not always identical and is forced to be aligned, where sometimes zero logits are padded to the end of the sequence. This suggests that clipped data may compose of useless zero frames. And the variant LGCCT with length scaled attention manages to generalize to the sequence with the length different from the training set.

Reviewer#1:

  1. The Conclusions part is far too short and not convincing. What about open issues and future studies? Provide additional feedback as well as source of inspiration for other authors.

Author Response: Thanks for your advice. Now the conclusion part involves the intuitional interpretation of the success of our LGCCT and the relationship between our design and the classic methods in deep learning community. Besides, we discuss our application of entropy and then encourage more deep learning researchers to pay more attention to fundamental application of entropy and information theory. The conclusion part is as follows.

Conclusion: In this paper, we propose LGCCT, a lightweight gated and crossed complementation Transformer for multimodal speech emotion recognition. Text encoder BiLSTM and audio encoder CNN-BiLSTM are utilized to obtain the modality feature expression respectively. At the heart of LGCCT, cross attention modules fuse the modality information with each other and the learnable gate control mechanism controls the information flow and stabilizes the training process. Moreover, we apply length scaling to the attention module to improve the generalization of transformer to various testing strings, which can be elaborated from a view of entropy invariance. In particular, the attention weight of the same token in the attention matrix is supposed to converge to the same value independent of the length. From the view of entropy, we can consider the uncertainty as the degree of attention focus. The attention scores can be consistent with the length of the input length just by multiplying the hyperparameter λ. At the top of the model, the fully connected forward network followed by the transformer encoder learns the mapping from modality-fused representation to emotion categories. In the experiments, we compare our model with baseline models on the benchmark dataset and further the ablation study. Our model achieves the balance between performance and the number of parameters with only 0.432M parameters. The results also show the effectiveness of each component, underlying the performance and lightweight of our model. And the length scale attention does help the model generalize to various sequence length under the experiment setting with different sequence lengths.

From the view of multimodal speech emotion recognition, our method has shown the balance between performance and the number of parameters. We attribute this to two factors. First, we adopt efficient modules in our network, such as cross-attention, gate control mechanism. In this way not only can the inter-modality information communicate and mingle with each other but the generated modality-fused information can also maintain the balance between the target modality and the source modality. Similar efficiency maintains within the process of feature extraction. Second, we keep the dimension of the hidden states low. Just as shown in Table 1, all the dimension are kept approximately 30, except for the input channels. In contrast, many previous works adopt BERT encoder [35] have to adjust the input channel to 512 [15,16], far more than ours. It is noteworthy that we don’t apply any other down-sampling after feature extraction stage, keeping the embedding dimension low and the sequence length the same. This design is similar to the classic ideas in computer vision, namely maintaining large activation maps while decreasing the quantity of parameters [57]. For a wide neural network, not all of the information contained in high-dimensional vectors are useful [58] and thus our LGCCT is designed as a narrower network so that the information can be more compact.

From the view of information theory, our entropy-based LGCCT variant is capable of generalizing to testing sequence with various lengths. We consider the degree of attention focus as the uncertainty and let the same token converge to the same value independent of the length by multiplying the predefined constant λ. Since the attention matrix is computed by the learnable parameters, the model is supposed to learn the value ideally. However, our experiment shows that the model with length scaled perform more stably while the model without length scaled fluctuate in performance. This phenomenon indicates that the perspective of entropy really works and the inductive bias [59] can help model find better solutions. Actually, the idea of entropy is widely applied to deep learning methods. One of the most typical applications is cross entropy. This criterion serves as a loss function to compute the loss between input and target especially when handling a classification problem with multiple classes [60]. More generally, other classic perspectives in information theory have been used in deep learning methods [47,61,62].

In the future, we will design more effective gate mechanism, following some gate units such as LSTM cell and GRU gates [63]. Besides, other modalities like video can be considered so that the tri-modal emotion recognition network can be developed for application in realistic scenarios. The effectiveness of length scaled attention for multimodal emotion recognition may shed light on the wider usage of entropy as well as information theory in deep learning community.

Reviewer#1:

  1. The References section, taking into account the topic of this paper, is far too short. It requires extensive extention.

Author Response: Thanks for your advice, we have extended the references from 35 to 63.

We thank the reviewers for their guidance on our content, which has been very helpful in improving the quality of our articles, thank you!

Reviewer 2 Report

I found the manuscript a very interesting one to read. The application area was one I found quite novel and very interesting. The AI method employed seems suitable and one that appears to work well.

Author Response

Thank you for your review! We will continue to improve the quality of our articles.